# Discriminative Power of Geometric Parameters of Different Cultivars of Sour Cherry Pits Determined Using Machine Learning

**Ewa Ropelewska [1,\*], Kadir Sabanci [2] and Muhammet Fatih Aslan [2]**

[1]  Fruit and Vegetable Storage and Processing Department, The National Institute of Horticultural Research, Konstytucji 3 Maja 1/3, 96-100 Skierniewice, Poland

[2]  Department of Electrical and Electronics Engineering, Karamanoglu Mehmetbey University, Karaman 70100, Turkey; kadirsabanci@kmu.edu.tr (K.S.); mfatihaslan@kmu.edu.tr (M.F.A.)

\*  Correspondence: ewa.ropelewska@inhort.pl

**Abstract:** The aim of this study was to develop models based on linear dimensions or shape factors, and the sets of combined linear dimensions and shape factors for discrimination of sour cherry pits of different cultivars ('Debreceni botermo', 'Łutówka', 'Nefris', 'Kelleris'). The geometric parameters were calculated using image processing. The pits of different sour cherry cultivars statistically significantly differed in terms of selected dimensions and shape factors. The discriminative models built based on linear dimensions produced average accuracies of up to 95% for distinguishing the pit cultivars in the case of 'Nefris' vs. 'Kelleris' and 72% for all four cultivars. The average accuracies for the discriminative models built based on shape factors were up to 95% for the 'Nefris' and 'Kelleris' pits and 73% for four cultivars. The models combining the linear dimensions and shape factors produced accuracies reaching 96% for the 'Nefris' vs. 'Kelleris' pits and 75% for all cultivars. The geometric parameters with high discriminative power may be used for distinguishing different cultivars of sour cherry pits. It can be of great importance for practical applications. It may allow avoiding the adulteration and mixing of different cultivars.

**Keywords:** sour cherry cultivars; pit images; linear dimensions; shape factors; discrimination

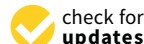



## 1. Introduction

Sour (tart) cherry (*Prunus cerasus* L.) is one of the two main species from the Prunus genus, besides sweet cherry (*Prunus avium* L.), with fruits globally traded. These fruit crops have been used by humans since 5000–4000 BCE, which was determined based on cherry pits from archaeological sites. Nowadays, there are many sour cherry cultivars. Due to the health benefits of cherries, tree crop cultivation should increase, and processing technology should be improved [1]. The cherry fruit has low caloric content and significant amounts of nutrients and bioactive components, e.g., polyphenols, fiber, vitamin C, carotenoids, potassium, as well as melatonin, serotonin, and tryptophan. A small number of sour cherries is consumed fresh. Up to 97% of fruits are processed mainly for cooking or baking [2]. Before processing, cherries are usually accurately pitted, as the unintended pits in processed cherry products may be a major concern for consumers (potential for injury) and processors (litigation) [3]. The pit of cherry fruit accounts for 6.30% by weight or even 7–15% of the whole fruit and it consists of the shell (75–80%) and kernel (20–25%) [4,5]. The very hard shell contains sclerenchyma and fiber matters. The kernel contains dietary proteins and fiber, and it has antimicrobial and antioxidant activities. The kernels may be used for the production of oils for the pharmaceutical, perfume and cosmetic industries or the production of biodiesel [4]. Additionally, cherry pit biomass may be potentially used for conversion into biochar for water remediation. This biomass may be also cofired with coal for the generation of electricity. The cherry pit biochar may be applied as catalyst

supports, alkaline-functionalized gas adsorbents, electrode materials, or soil amendments for greenhouse crop production [6–11]. However, pits are still an important waste disposal problem for the processing industry [4]. The traditional waste disposal should be replaced by greener ways of cherry pit biomass application [11].

Depending on the extraction procedure and roasting process, the nutrients may pass from the sour cherry kernels into the oil at different percentages [12]. The sour cherry cultivar may also influence the oil content of the kernel that is about 17–36% [5]. The cultivar of cherry kernel also has a great effect on lipophilic bioactive compounds, e.g., sterols, essential fatty acids, tocopherols, tocochromanols, squalene, carotenoids [5,13]. Due to the dependence of the chemical properties of sour cherry kernels on the cultivar, correct cultivar recognition may be important in practice. The processing of cherry kernels may require a uniform sample of kernels with the same characteristics. Some cultivars with certain chemical properties may be more desirable for processing than others. Therefore, there may be a need for authentication to avoid adulteration and mixing different cultivars.

The application of machine learning may be useful for plant research. Machine learning as a sub-class of artificial intelligence is an important topic in the computer field. Currently, researchers strive to increase the precision of algorithms and the intelligence of machines. Learning became a significant part of machines. Due to computer vision, which is a domain of machine learning, machines can be trained for processing, analyzing, and recognizing visual data [14]. Machine learning is intended to enable machines to learn using the available data and make predictions. The learning of computers automatically by themselves without human intervention may be important for precise prediction [15]. The prediction models developed using machine learning and artificial intelligence can provide promising and accurate results. The models based on artificial intelligence can learn from existing data and then predict even nonlinear phenomena related to, e.g., prediction of food production, crop yield, or identification of the number of immature fruits [16]. The application of machine learning in modern agriculture is important due to the increasing call for food, the necessity for increasing the effectiveness of agricultural practices and decreasing the environmental burden. Machine learning ensures an increase in computational power compared to conventional techniques of data processing, which can be incapable of extracting all necessary information from field data and thus meeting the growing demands of smart farming [17]. Machine learning focused on the detection of disease, species, and weeds in crops, the prediction of crop yield and soil parameters, and the classification of crop images to evaluate the plant quality and yield can be one of the key components of the agricultural revolution [18].

In the case of the seed industry, machine learning may be important for the production, correct cultivar identification, identification of contaminations, and quality control. The use of machine vision techniques can result in more accurate and faster classification results compared to the manual inspection performed by specialists based on the color and morphological features of seeds [19]. Machine learning caused significant advances in seed research by providing decision-making support and facilitating the development of robust approaches in the seed industry [20]. The usefulness of the application of machine learning for seed classification was reported in the available literature. The machine learning models were built based on various image features. In the case of cultivar discrimination of fruit seeds or pits and stones, the high efficiency of models based on texture parameters was reported for pepper seeds [21], apple seeds [22], peach seeds and stones [23], sour cherry pits [24], and sweet cherry pits [25]. Furthermore, the geometric features proved to be useful for the pit or stone discrimination for different cultivars of apricot [26], plum [27–29], olive [30], jujube [31], and sweet cherry [25]. However, in the present study, extensive research using dozens of geometric parameters, including linear dimensions and shape factors, was performed for the first time to discriminate sour cherry pits 'Debreceni botermo', 'Łutówka', 'Nefris', 'Kelleris' using different classifiers (machine learning algorithms). The innovative models based on the sets of selected linear dimensions,

shape factors, and combined linear dimensions and shape factors were developed. This approach to distinguishing cultivars of sour cherry pits is original.

The aim of this study was to develop discriminative models based on geometric features including linear dimensions and, separately, shape factors, as well as the combination of linear dimensions and shape factors for the discrimination of the sour cherry pits of different cultivars. The discriminative power of geometric parameters for distinguishing the pairs of cultivars and all four cultivars was compared.

## 2. Materials and Methods

### 2.1. Materials

The pits of sour cherries 'Debreceni botermo', 'Łutówka', 'Nefris', and 'Kelleris' were used in the research. The cherries were collected from the Experimental Orchard of the National Institute of Horticultural Research in Dąbrowice near Skierniewice (Poland). The pits were manually extracted from the fruits. For each cultivar, 'Debreceni botermo', 'Łutówka', 'Nefris', 'Kelleris', two hundred pits were sampled, washed, cleaned and air-dried.

### 2.2. Image Analysis

The pits were imaged using a flatbed scanner. The sour cherry pits were scanned on a black background at the 1200 dpi resolution and the pit images were saved in TIFF. The images of sour cherry pits were analyzed with the use of Mazda software (Łódź University of Technology, Institute of Electronics, Poland) [32]. For each pit, the region of interest (ROI) including the whole pit was determined. A caliper image was used for the calibration. Then, for each pit with overlaid ROI, the geometric parameters were computed. Among the linear dimensions, the following features were determined: length ($L$); width ($S$); length of the skeletonized object ($L_{sz}$); area of circumscribing ellipse on the object ($FE$); maximal length of the ellipse axis on the object ($L_{maxE}$); minimal length of the ellipse axis on the object ($L_{minE}$); area of circumscribing circle ($Fd_2$); radius of circumscribing circle ($D_2$); profile specific perimeter ($Ul$); Martin's maximal radius ($M_{max}$); Martin's minimal radius ($M_{min}$); vertical Feret diameter ($F_v$); convex perimeter ($U_w$); object boundary specific perimeter ($U_g$); equivalent circular area diameter ($S_{pol}$); total object specific area ($F_t$); horizontal Feret diameter ($F_h$); maximal Feret diameter ($F_{max}$); minimal Feret diameter ($F_{min}$); Martin's average radius ($M_{aver}$). The calculated shape factors included: elliptic shape factor ($W_1$); circular shape factor ($W_2$); circularity ($W_3$); folding factor ($W_4$); mean thickness factor ($W_5$); elongation and irregularity ratio ($W_7$); rectangular aspect ratio ($W_8$); area ratio ($W_9$); radius ratio ($W_{10}$); diameter range ($W_{11}$); roundness ($(4\,\pi\,F)/(\pi\,S_{max}^2)$) ($W_{12}$); roundness ($S_{max}/F$) ($W_{13}$); roundness ($F/S_{max}^3$) ($W_{14}$); roundness ($4F/(\pi\,S_{min}\,S_{max})$) ($W_{15}$); standard deviation of all radii ($SigR$); Haralick ratio ($R_H$); Blair–Bliss ratio ($R_B$); Malinowska ratio ($R_M$); Feret ratio ($F_h/F_v$) ($R_F$); Feret ratio ($F_{max}/F_{min}$) ($R_{Ff}$); circularity ($R_{c1}/R_{c2}$) ($R_c$); circularity ($2\sqrt{(F/\pi)}$) ($R_{c1}$); circularity ($U_g/\pi$) ($R_{c2}$).

### 2.3. Statistical Analysis

The mean values of the linear dimensions and shape factors of the pits of sour cherries 'Debreceni botermo', 'Łutówka', 'Nefris', and 'Kelleris' were compared to determine the differences in parameters between sour cherry cultivars. The STATISTICA (StatSoft Inc., Tulsa, OK, USA) software program was used at a significance level of $p \leq 0.05$. The normality of the distribution was checked using Kolmogorov–Smirnov, Lilliefors and Shapiro–Wilk tests. The Newman–Keuls test was used for the comparison of the means. The homogenous groups of sour cherry pits had no statistically significant differences in the geometric parameters and were indicated by the same letters in columns. The separate groups in terms of linear dimensions or shape factors with statistically significant differences were indicated by different letters in columns.

The usefulness of geometric parameters including linear dimensions and shape factors for distinguishing the pits of sour cherries belonging to different cultivars was analyzed using the WEKA (Machine Learning Group, University of Waikato) application [33]. In the

first step of the analysis, the discriminative models were built based on linear dimensions. In the next step, the models based on shape factors were developed. Then, the discriminative models were built based on datasets of the combined linear dimensions and shape factors. The discriminative models were developed separately for each pair of cultivars and all four cultivars. The attribute selection to choose the parameters with the highest discriminative power was carried out using the Best First with the correlation-based feature selection (CFS) subset evaluator, the Ranker method with the Info Gain attribute evaluator, the Ranker method with the OneR attribute evaluator, the Genetic Search method with the CFS subset evaluator. The criterion for evaluating the usefulness of datasets selected with the use of search methods was the highest correctness of discrimination. However, a great reduction in the number of parameters decreased the correctness of the discrimination and analyzes were performed with the exclusion of only a few attributes. The datasets were manually split into a training (70%) and test set (30%). The application of a separate test set that was not used for training ensured the objectivity of the results. The discrimination was performed using the classifiers (machine learning algorithms): NaiveBayes, BayesNet (from the group of Bayes), JRip, PART (Rules), J48, RandomTree (decision trees), Logistic, MultilayerPerceptron (Functions), MultiClassClassifier, FilteredClassifier (Meta), and IBk, KStar (Lazy) [34]. Based on preliminary observations, the highest classification accuracy for discriminative models was found for the Logistic method and the results obtained for this classifier are shown in this paper. The results are presented as confusion matrices and average accuracies (rounded to integers), as well as the values of the true positive (TP) rate, precision, F-measure, receiver operating characteristic (ROC) area and precision–recall (PRC) area calculated using the Weka application based on the formulas:

$$\text{TP Rate} = TP/(TP + FN) \tag{1}$$

$$\text{Precision} = TP/(TP + FP) \tag{2}$$

$$\text{F-Measure} = 2 \times ((\text{Precision} \times \text{Recall})/(\text{Precision} + \text{Recall})) \tag{3}$$

$$\text{Recall} = TP/(TP + FN) \tag{4}$$

where TP is true positive; FP is false positive; FN is false negative.

## 3. Results and Discussion

The linear dimensions of 'Debreceni botermo', 'Łutówka', 'Nefris', and 'Kelleris' cherry pits were compared to determine the differences in the mean values between cultivars (Table 1). All four pit cultivars were different in the terms of their basic linear dimensions, such as length ($L$) and width ($S$). Each cultivar formed a separate homogenous group. The 'Kelleris' pits were characterized by the highest mean values of the parameter $L$ equal to 12.14 mm. Subsequently, the length of the 'Nefris', 'Łutówka', and 'Debreceni botermo' pits was 11.80 mm, 11.54 mm, and 11.33 mm, respectively. The mean value of parameter $S$ was the highest for the 'Nefris' pits (10.49 mm), followed by 'Debreceni botermo' (10.09 mm), 'Łutówka' (9.87 mm), and 'Kelleris' (9.49 mm). The four homogenous groups were also determined in the case of the length of the skeletonized object ($L_{sz}$), Martin's minimal radius ($M_{min}$), and minimal Feret diameter ($F_{min}$). In the case of these parameters, the 'Nefris' pits were characterized by the highest values ($L_{sz}$—174.71 mm, $M_{min}$—4.92 mm, $F_{min}$—10.29 mm) and the 'Kelleris' pits had the lowest values ($L_{sz}$—125.55 mm, $M_{min}$—4.45 mm, $F_{min}$—9.32 mm). In the case of many parameters ($U_w$, $U_g$, $S_{pol}$, $F_t$, $F_h$, $M_{aver}$), the 'Debreceni botermo', 'Łutówka', and 'Kelleris' pits were in one homogenous group and the 'Nefris' pits formed the second homogenous group with a statistically significantly different mean value.

**Table 1.** Comparison of the mean values of linear dimensions of 'Debreceni botermo', 'Łutówka', 'Nefris', and 'Kelleris' cherry pits.

| Parameter | Cultivar | | | |
|---|---|---|---|---|
| | 'Debreceni Botermo' | 'Łutówka' | 'Nefris' | 'Kelleris' |
| $L$ (mm) | 11.33 [a] | 11.54 [b] | 11.80 [c] | 12.14 [d] |
| $S$ (mm) | 10.09 [c] | 9.87 [b] | 10.49 [d] | 9.49 [a] |
| $L_{sz}$ (mm) | 141.68 [b] | 150.88 [c] | 174.71 [d] | 125.55 [a] |
| $FE$ (mm$^2$) | 116.98 [a] | 114.59 [a] | 119.62 [b] | 121.63 [b] |
| $L_{maxE}$ (mm) | 12.42 [a] | 12.23 [b] | 12.42 [a] | 12.68 [c] |
| $L_{minE}$ (mm) | 11.96 [a] | 11.90 [a] | 12.23 [b] | 12.18 [b] |
| $Fd_2$ (mm$^2$) | 120.41 [a] | 117.07 [b] | 121.14 [a] | 125.20 [c] |
| $D_2$ (mm) | 6.18 [a] | 6.10 [b] | 6.20 [a] | 6.30 [c] |
| $Ul$ (mm) | 100.37 [a] | 101.16 [a] | 105.06 [b] | 100.83 [a] |
| $M_{max}$ (mm) | 6.27 [a] | 6.21 [b] | 6.31 [a] | 6.39 [c] |
| $M_{min}$ (mm) | 4.67 [c] | 4.58 [b] | 4.92 [d] | 4.45 [a] |
| $F_v$ (mm) | 10.69 [bc] | 10.44 [a] | 10.86 [c] | 10.59 [ab] |
| $U_w$ (mm) | 34.44 [a] | 34.19 [a] | 35.60 [b] | 34.52 [a] |
| $U_g$ (mm) | 100.45 [a] | 101.54 [a] | 105.19 [b] | 101.50 [a] |
| $S_{pol}$ (mm) | 10.70 [a] | 10.62 [a] | 11.12 [b] | 10.67 [a] |
| $F_t$ (mm$^2$) | 90.20 [a] | 88.74 [a] | 97.31 [b] | 89.57 [a] |
| $F_h$ (mm) | 11.09 [a] | 11.13 [a] | 11.59 [b] | 11.23 [a] |
| $F_{max}$ (mm) | 12.34 [a] | 12.15 [b] | 12.35 [a] | 12.58 [c] |
| $F_{min}$ (mm) | 9.78 [c] | 9.63 [b] | 10.29 [d] | 9.32 [a] |
| $M_{aver}$ (mm) | 5.36 [a] | 5.32 [a] | 5.56 [b] | 5.35 [a] |

$L$—length; $S$—width; $L_{sz}$—length of the skeletonized object; $FE$—area of circumscribing ellipse on the object; $L_{maxE}$—maximal length of the ellipse axis on the object; $L_{minE}$—minimal length of the ellipse axis on the object; $Fd_2$—area of circumscribing circle; $D_2$—radius of circumscribing circle; $Ul$—profile specific perimeter; $M_{max}$—Martin's maximal radius; $M_{min}$—Martin's minimal radius; $F_v$—vertical Feret diameter; $U_w$—convex perimeter; $U_g$—object boundary specific perimeter; $S_{pol}$—equivalent circular area diameter; $F_t$—total object specific area; $F_h$—horizontal Feret diameter; $F_{max}$—maximal Feret diameter; $F_{min}$—minimal Feret diameter; $M_{aver}$—Martin's average radius. [a,b,c,d]—the same letters in rows denote no statistical differences between samples.

The mean values of the shape factors of 'Debreceni botermo', 'Łutówka', 'Nefris', and 'Kelleris' cherry pits are presented in Table 2. In terms of some parameters, such as mean thickness factor ($W_5$), compactness ($W_6$), area ratio ($W_9$), roundness ($W_{12}$) and ($W_{14}$), Malinowska ratio ($R_M$), and circularity ($R_c$), the pits were statistically significantly different, and each cultivar formed a separate homogenous group. For one parameter, Feret ratio ($R_F$), the pits belonging to all cultivars were in one homogenous group with no statistically significant differences between the mean values. In the case of most shape factors, elliptic shape factor ($W_1$), circular shape factor ($W_2$), circularity ($W_3$), elongation and irregularity ratio ($W_7$), rectangular aspect ratio ($W_8$), radius ratio ($W_{10}$), diameter range ($W_{11}$), roundness ($W_{13}$) and ($W_{15}$), standard deviation of all radii ($SigR$), Haralick ratio ($R_H$), Blair–Bliss ratio ($R_B$), and Feret ratio ($R_{Ff}$), three homogenous groups were formed, and in most cases ($W_7$, $W_{10}$, $W_{11}$, $W_{13}$, $SigR$, $R_H$, $R_{Ff}$), the 'Debreceni botermo' and 'Łutówka' pits were in one group.

**Table 2.** Comparison of the mean values of shape factors of 'Debreceni botermo', 'Łutówka', 'Nefris', and 'Kelleris' cherry pits.

| Parameter | Cultivar | | | |
|---|---|---|---|---|
| | 'Debreceni Botermo' | 'Łutówka' | 'Nefris' | 'Kelleris' |
| $W_1$ (-) | 1.04 [a] | 1.03 [b] | 1.02 [c] | 1.05 [a] |
| $W_2$ (-) | 0.11 [c] | 0.11 [b] | 0.11 [a] | 0.11 [a] |
| $W_3$ (-) | 111.91 [b] | 115.58 [c] | 113.67 [a] | 113.83 [a] |
| $W_4$ (-) | 2.91 [a] | 2.96 [b] | 2.95 [b] | 2.92 [a] |
| $W_5$ (-) | 0.67 [c] | 0.61 [b] | 0.57 [a] | 0.75 [d] |
| $W_6$ (-) | 0.09 [b] | 0.09 [d] | 0.09 [a] | 0.09 [c] |
| $W_7$ (-) | 1.30 [a] | 1.30 [a] | 1.24 [b] | 1.38 [c] |
| $W_8$ (-) | 0.89 [a] | 0.86 [c] | 0.89 [a] | 0.78 [b] |
| $W_9$ (-) | 1.27 [a] | 1.28 [c] | 1.27 [b] | 1.29 [d] |
| $W_{10}$ (-) | 0.75 [a] | 0.74 [a] | 0.78 [c] | 0.70 [b] |
| $W_{11}$ (-) | 2.57 [a] | 2.52 [a] | 2.52 [b] | 3.27 [c] |
| $W_{12}$ (-) | 2.37 [b] | 2.41 [c] | 2.55 [d] | 2.26 [a] |
| $W_{13}$ (-) | 0.14 [a] | 0.14 [a] | 0.13 [b] | 0.14 [c] |
| $W_{14}$ (-) | 0.05 [b] | 0.05 [c] | 0.05 [d] | 0.04 [a] |
| $W_{15}$ (-) | 0.95 [b] | 0.96 [c] | 0.97 [a] | 0.97 [a] |
| $SigR$ (-) | 204.90 [a] | 198.94 [a] | 133.71 [b] | 322.29 [c] |
| $R_H$ (-) | 1.00 [a] | 1.00 [a] | 1.00 [c] | 0.99 [b] |
| $R_B$ (-) | 9.35 [b] | 9.28 [ab] | 9.77 [c] | 9.24 [a] |
| $R_M$ (-) | 10.94 [a] | 11.17 [d] | 11.04 [b] | 11.11 [c] |
| $R_F$ (-) | 1.05 [a] | 1.08 [a] | 1.08 [a] | 1.08 [a] |
| $R_{Ff}$ (-) | 0.79 [a] | 0.79 [a] | 0.83 [c] | 0.74 [b] |
| $R_c$ (-) | 0.33 [d] | 0.33 [a] | 0.33 [c] | 0.33 [b] |
| $R_{c1}$ (-) | 10.70 [a] | 10.62 [a] | 11.12 [b] | 10.67 [a] |
| $R_{c2}$ (-) | 31.97 [a] | 32.32 [a] | 33.48 [b] | 32.31 [a] |

$W_1$—elliptic shape factor; $W_2$—circular shape factor; $W_3$—circularity; $W_4$—folding factor; $W_5$—mean thickness factor; $W_6$—compactness; $W_7$—elongation and irregularity ratio; $W_8$—rectangular aspect ratio; $W_9$—area ratio; $W_{10}$—radius ratio; $W_{11}$—diameter range; $W_{12}$—roundness $((4 \pi F)/(\pi S_{max}^2))$; $W_{13}$—roundness $(S_{max}/F)$; $W_{14}$—roundness $(F/S_{max}^3)$; $W_{15}$—roundness $(4F/(\pi S_{min} S_{max}))$; $SigR$—standard deviation of all radii; $R_H$—Haralick ratio; $R_B$—Blair–Bliss ratio; $R_M$—Malinowska ratio; $R_F$—Feret ratio $(F_h/F_v)$; $R_{Ff}$—Feret ratio $(F_{max}/F_{min})$; $R_c$—circularity $(R_{c1}/R_{c2})$; $R_{c1}$—circularity $(2\sqrt{(F/\pi)})$; $R_{c2}$—circularity $(U_g/\pi)$. [a,b,c,d]—the same letters in rows denote no statistical differences between samples.

In the first step of the discriminant analysis, the cherry pits were compared in pairs including two different cultivars. The results of the discrimination based on selected linear dimensions are presented in Table 3. The highest average accuracy of 95% was determined in the case of distinguishing between 'Nefris' and 'Kelleris' pits. The confusion matrix revealed that 95% of the pits belonging to 'Nefris' were correctly included in the class 'Nefris' and 5% incorrectly assigned to the class 'Kelleris', whereas 94% of 'Kelleris' pits were correctly included in the class 'Kelleris' and 6% were incorrectly included in the class 'Nefris'. For these pit cultivars, the values of the true positive (TP) rate ('Nefris'— 0.95, 'Kelleris'—0.94), precision ('Nefris'—0.94, 'Kelleris'—0.96), F-measure ('Nefris'—0.94, 'Kelleris'—0.95), ROC (Receiver Operating Characteristic) Area ('Nefris'—0.97, 'Kelleris'— 0.97) and precision–recall (PRC) area ('Nefris'—0.95, 'Kelleris'—0.95) were the highest. It may indicate that the 'Nefris' and 'Kelleris' pits were the most different in terms of linear dimensions. It confirmed the results of the comparison of the mean values of linear dimensions (Table 1) that indicated that for most parameters, the 'Nefris' and 'Kelleris' pits were not in one homogenous group and in some cases formed two of the most distant groups. The lowest average accuracies were observed for the discrimination of the pits of cherry 'Łutówka' vs. 'Nefris' (78%) and 'Debreceni botermo' vs. 'Łutówka' (84%). In these cases, the linear dimensions had the lowest discriminative power. The 'Łutówka' and 'Nefris' pits, as well as those of 'Debreceni botermo' and 'Łutówka' were the most similar in terms of length. The difference in length between the 'Łutówka' and 'Nefris' pits

was 0.26 mm and the difference between the 'Debreceni botermo' and 'Łutówka' pits was equal to 0.21 mm (Table 1). In the case of other pairs of cherry pits, an average accuracy of 90% was found for distinguishing 'Debreceni botermo' vs. 'Kelleris', 87% for 'Debreceni botermo' vs. 'Nefris' and 'Łutówka' vs. 'Kelleris' (Table 3).

**Table 3.** The discrimination performance for the pair comparison of the pits of cherry 'Debreceni botermo', 'Łutówka', 'Nefris', and 'Kelleris' based on selected linear dimensions.

| Pair Comparison | Predicted Class (%) | | Actual Class | Average Accuracy (%) | TP Rate | Precision | F-Measure | ROC Area | PRC Area |
|---|---|---|---|---|---|---|---|---|---|
| 'Debreceni botermo' vs. 'Łutówka' | 'Debreceni botermo' | 'Łutówka' | | | | | | | |
| | 85 | 15 | 'Debreceni botermo' | 84 | 0.85 | 0.86 | 0.86 | 0.91 | 0.92 |
| | 16 | 84 | 'Łutówka' | | 0.84 | 0.82 | 0.83 | 0.91 | 0.87 |
| 'Debreceni botermo' vs. 'Nefris' | 'Debreceni botermo' | 'Nefris' | | | | | | | |
| | 87 | 13 | 'Debreceni botermo' | 87 | 0.87 | 0.90 | 0.88 | 0.93 | 0.94 |
| | 13 | 87 | 'Nefris' | | 0.87 | 0.84 | 0.85 | 0.93 | 0.92 |
| 'Debreceni botermo' vs. 'Kelleris' | 'Debreceni botermo' | 'Kelleris' | | | | | | | |
| | 92 | 8 | 'Debreceni botermo' | 90 | 0.92 | 0.91 | 0.91 | 0.95 | 0.93 |
| | 11 | 89 | 'Kelleris' | | 0.89 | 0.90 | 0.89 | 0.95 | 0.93 |
| 'Łutówka' vs. 'Nefris' | 'Łutówka' | 'Nefris' | | | | | | | |
| | 78 | 22 | 'Łutówka' | 78 | 0.78 | 0.79 | 0.78 | 0.85 | 0.85 |
| | 22 | 78 | 'Nefris' | | 0.78 | 0.76 | 0.77 | 0.85 | 0.82 |
| 'Łutówka' vs. 'Kelleris' | 'Łutówka' | 'Kelleris' | | | | | | | |
| | 87 | 13 | 'Łutówka' | 87 | 0.87 | 0.86 | 0.87 | 0.92 | 0.91 |
| | 13 | 87 | 'Kelleris' | | 0.87 | 0.87 | 0.87 | 0.92 | 0.92 |
| 'Nefris' vs. 'Kelleris' | 'Nefris' | 'Kelleris' | | | | | | | |
| | 95 | 5 | 'Nefris' | 95 | 0.95 | 0.94 | 0.94 | 0.97 | 0.95 |
| | 6 | 94 | 'Kelleris' | | 0.94 | 0.96 | 0.95 | 0.97 | 0.95 |

TP Rate—true positive rate; ROC Area—receiver operating characteristic area; PRC Area—precision–recall area.

The results of discrimination of the pairs of pits of cherry 'Debreceni botermo', 'Łutówka', 'Nefris', 'Kelleris' based on shape factors are shown in Table 4. The tendency was similar to the results of discriminative models built based on linear dimensions (Table 3). In both cases, the 'Nefris' and 'Kelleris' pits were characterized by the highest average discrimination accuracy of 95% (Tables 3 and 4). The sour cherry pits of 'Łutówka' vs. 'Nefris' (78%) (Tables 3 and 4) and 'Debreceni botermo' vs. 'Łutówka' (84% (Table 3), 85% (Table 4)) had the lowest average accuracies. The other discriminative models built based on shape factors produced average accuracies of 92% for 'Debreceni botermo' vs. 'Kelleris' pits, 88% for 'Debreceni botermo' vs. 'Nefris' pits, 87% for 'Łutówka' vs. 'Kelleris' pits (Table 4). It indicated that the accuracies for models built based on shape factors (Table 4) were slightly higher than models built based on linear dimensions (Table 3).

**Table 4.** The discrimination performance for the pair comparison of the pits of cherry 'Debreceni botermo', 'Łutówka', 'Nefris', and 'Kelleris' based on selected shape factors.

| Pair Comparison | Predicted Class (%) | | Actual Class | Average Accuracy (%) | TP Rate | Precision | F-Measure | ROC Area | PRC Area |
|---|---|---|---|---|---|---|---|---|---|
| 'Debreceni botermo' vs. 'Łutówka' | 'Debreceni botermo' | 'Łutówka' | | | | | | | |
| | 87 | 13 | 'Debreceni botermo' | 85 | 0.87 | 0.86 | 0.86 | 0.91 | 0.93 |
| | 18 | 82 | 'Łutówka' | | 0.82 | 0.84 | 0.83 | 0.91 | 0.87 |
| 'Debreceni botermo' vs. 'Nefris' | 'Debreceni botermo' | 'Nefris' | | | | | | | |
| | 89 | 11 | 'Debreceni botermo' | 88 | 0.89 | 0.90 | 0.90 | 0.94 | 0.93 |
| | 13 | 87 | 'Nefris' | | 0.87 | 0.86 | 0.86 | 0.94 | 0.88 |
| 'Debreceni botermo' vs. 'Kelleris' | 'Debreceni botermo' | 'Kelleris' | | | | | | | |
| | 92 | 8 | 'Debreceni botermo' | 92 | 0.92 | 0.93 | 0.92 | 0.96 | 0.96 |
| | 8 | 92 | 'Kelleris' | | 0.92 | 0.90 | 0.91 | 0.96 | 0.92 |
| 'Łutówka' vs. 'Nefris' | 'Łutówka' | 'Nefris' | | | | | | | |
| | 77 | 23 | 'Łutówka' | 78 | 0.77 | 0.79 | 0.78 | 0.86 | 0.86 |
| | 21 | 79 | 'Nefris' | | 0.79 | 0.76 | 0.77 | 0.86 | 0.80 |
| 'Łutówka' vs. 'Kelleris' | 'Łutówka' | 'Kelleris' | | | | | | | |
| | 87 | 13 | 'Łutówka' | 87 | 0.87 | 0.86 | 0.87 | 0.94 | 0.94 |
| | 13 | 87 | 'Kelleris' | | 0.87 | 0.87 | 0.87 | 0.94 | 0.93 |
| 'Nefris' vs. 'Kelleris' | 'Nefris' | 'Kelleris' | | | | | | | |
| | 95 | 5 | 'Nefris' | 95 | 0.95 | 0.95 | 0.95 | 0.98 | 0.96 |
| | 5 | 95 | 'Kelleris' | | 0.95 | 0.96 | 0.95 | 0.98 | 0.97 |

TP Rate—true positive rate; ROC Area—receiver operating characteristic area; PRC Area—precision–recall area.

The accuracies of discrimination based on selected combined linear dimensions and shape factors (Table 5) were higher than for the discrimination performed with shape factors (Table 4) and linear dimensions (Table 3). In the case of models built based on sets of combined linear dimensions and shape factors (Table 5), the average accuracy reached 96% for distinguishing 'Nefris' and 'Kelleris'. It is 1% higher than for the discrimination of the 'Nefris' and 'Kelleris' pits for models built based on linear dimensions (95%, Table 3) and shape factors (95%, Table 4). In addition, the lowest accuracy of 79%, determined based on combined linear dimensions and shape factors for 'Łutówka' vs. 'Nefris' pits (Table 5), was 1% higher than for the model based on linear dimensions (78%, Table 3) and shape factors (78%, Table 4) for the discrimination of the 'Łutówka' and 'Nefris' pits. Furthermore, the discrimination accuracies for all other pairs of cherry pits based on combined linear dimensions and shape factors (Table 5) increased and were equal to 86% for 'Debreceni botermo' vs. 'Łutówka', 89% for 'Debreceni botermo' vs. 'Nefris', 93% for 'Debreceni botermo' vs. 'Kelleris', and 90% for 'Łutówka' vs. 'Kelleris'.

The performance of the discrimination for all four cultivars was compared for the models built separately for linear dimensions, shape factors and combined linear dimensions and shape factors (Table 6). The average accuracy of 75% was the highest for discriminative models including combined linear dimensions and shape factors. In this analysis, the pits 'Debreceni botermo' and 'Kelleris' were characterized by an accuracy of 82%. The correctness of 76% was determined for the pits 'Nefris' and 59% for the pits 'Łutówka'. The least incorrectly classified cases were between the pits 'Nefris' and 'Kelleris', and the most incorrectly classified cases were between the pits 'Łutówka' and 'Nefris'. The discriminative models built based on shape factors produced an accuracy of 73%. The lowest average accuracy of discrimination of four cherry cultivars was observed for models built based on linear dimensions (72%). It indicated that combined linear dimensions and shape factors had the highest discriminative power for distinguishing the cherry pits belonging to different cultivars, and the discriminative power of linear dimensions was the lowest.

**Table 5.** The discrimination performance for the pair comparison of the pits of cherry 'Debreceni botermo', 'Łutówka', 'Nefris', and 'Kelleris' based on selected combined linear dimensions and shape factors.

| Pair Comparison | Predicted Class (%) | | Actual Class | Average Accuracy (%) | TP Rate | Precision | F-Measure | ROC Area | PRC Area |
|---|---|---|---|---|---|---|---|---|---|
| 'Debreceni botermo' vs. 'Łutówka' | 'Debreceni botermo' | 'Łutówka' | | | | | | | |
| | 87 | 13 | 'Debreceni botermo' | 86 | 0.87 | 0.88 | 0.87 | 0.92 | 0.92 |
| | 15 | 85 | 'Łutówka' | | 0.85 | 0.84 | 0.85 | 0.92 | 0.89 |
| 'Debreceni botermo' vs. 'Nefris' | 'Debreceni botermo' | 'Nefris' | | | | | | | |
| | 89 | 11 | 'Debreceni botermo' | 89 | 0.89 | 0.91 | 0.90 | 0.94 | 0.95 |
| | 12 | 88 | 'Nefris' | | 0.88 | 0.85 | 0.87 | 0.94 | 0.90 |
| 'Debreceni botermo' vs. 'Kelleris' | 'Debreceni botermo' | 'Kelleris' | | | | | | | |
| | 92 | 8 | 'Debreceni botermo' | 93 | 0.92 | 0.94 | 0.93 | 0.97 | 0.97 |
| | 7 | 93 | 'Kelleris' | | 0.93 | 0.91 | 0.92 | 0.97 | 0.96 |
| 'Łutówka' vs. 'Nefris' | 'Łutówka' | 'Nefris' | | | | | | | |
| | 79 | 21 | 'Łutówka' | 79 | 0.79 | 0.80 | 0.80 | 0.85 | 0.86 |
| | 21 | 79 | 'Nefris' | | 0.79 | 0.77 | 0.78 | 0.85 | 0.79 |
| 'Łutówka' vs. 'Kelleris' | 'Łutówka' | 'Kelleris' | | | | | | | |
| | 90 | 10 | 'Łutówka' | 90 | 0.90 | 0.89 | 0.90 | 0.94 | 0.92 |
| | 10 | 90 | 'Kelleris' | | 0.90 | 0.90 | 0.90 | 0.94 | 0.91 |
| 'Nefris' vs. 'Kelleris' | 'Nefris' | 'Kelleris' | | | | | | | |
| | 96 | 4 | 'Nefris' | 96 | 0.96 | 0.96 | 0.96 | 0.99 | 0.99 |
| | 4 | 96 | 'Kelleris' | | 0.96 | 0.96 | 0.96 | 0.98 | 0.98 |

TP Rate—true positive rate; ROC Area—receiver operating characteristic area; PRC Area—precision–recall area.

**Table 6.** The performance of discrimination of the pits of cherry 'Debreceni botermo', 'Łutówka', 'Nefris', and 'Kelleris' based on selected geometric parameters.

| Predicted Class (%) | | | | Actual Class | Average Accuracy (%) | TP Rate | Precision | F-Measure | ROC Area | PRC Area |
|---|---|---|---|---|---|---|---|---|---|---|
| | | | | Linear dimensions | | | | | | |
| 'Debreceni botermo' | 'Łutówka' | 'Nefris' | 'Kelleris' | | | | | | | |
| 76 | 9 | 7 | 8 | 'Debreceni botermo' | | 0.76 | 0.76 | 0.76 | 0.91 | 0.81 |
| 13 | 55 | 19 | 13 | 'Łutówka' | 72 | 0.55 | 0.60 | 0.57 | 0.82 | 0.58 |
| 11 | 17 | 71 | 1 | 'Nefris' | | 0.71 | 0.70 | 0.70 | 0.91 | 0.75 |
| 5 | 10 | 1 | 84 | 'Kelleris' | | 0.84 | 0.79 | 0.82 | 0.95 | 0.87 |
| | | | | Shape factors | | | | | | |
| 'Debreceni botermo' | 'Łutówka' | 'Nefris' | 'Kelleris' | | | | | | | |
| 79 | 9 | 7 | 5 | 'Debreceni botermo' | | 0.79 | 0.78 | 0.78 | 0.92 | 0.85 |
| 13 | 54 | 20 | 13 | 'Łutówka' | 73 | 0.54 | 0.59 | 0.56 | 0.83 | 0.60 |
| 8 | 18 | 73 | 1 | 'Nefris' | | 0.73 | 0.69 | 0.71 | 0.93 | 0.77 |
| 6 | 10 | 0 | 84 | 'Kelleris' | | 0.84 | 0.82 | 0.83 | 0.96 | 0.88 |
| | | | | Linear dimensions + shape factors | | | | | | |
| 'Debreceni botermo' | 'Łutówka' | 'Nefris' | 'Kelleris' | | | | | | | |
| 82 | 6 | 7 | 5 | 'Debreceni botermo' | | 0.82 | 0.82 | 0.82 | 0.93 | 0.84 |
| 9 | 59 | 20 | 12 | 'Łutówka' | 75 | 0.59 | 0.64 | 0.61 | 0.82 | 0.59 |
| 7 | 16 | 76 | 1 | 'Nefris' | | 0.76 | 0.70 | 0.73 | 0.93 | 0.77 |
| 7 | 10 | 1 | 82 | 'Kelleris' | | 0.82 | 0.81 | 0.81 | 0.95 | 0.88 |

TP Rate—true positive rate; ROC Area—receiver operating characteristic area; PRC Area—precision–recall area.

The results of the studies revealed the usefulness of the geometric parameters for the discrimination of different cultivars of sour cherry pits. Both linear dimensions and shape factors had a high discriminative power. However, the models built based on combined linear dimensions and shape factors provided the highest results, equal to 96%, for the discrimination of two pit cultivars and 75% for four pit cultivars. The results obtained by Ropelewska [24] indicated that the textures had even higher discriminative power for the discrimination of the pits of different sour cherry cultivars. The pairs of cultivars were discriminated with an average accuracy of up to 100%, whereas, for the discrimination of four cultivars, the correctness of up to 96.25% was achieved. Ropelewska [25] reported that for sweet cherry pits as well, the discrimination accuracies for models built based on textural features (up to 100% for two pit cultivars and 95% for three cultivars) were higher than for geometric parameters (up to 99% for two cultivars and 95% for three cultivars). Additionally, Ropelewska [25] found that the models combining geometric and textural parameters provided the highest accuracies of up to 100% for two cultivars and 98% for three pit cultivars. The results of cultivar discrimination of sour cherry pits based on geometric parameters presented in this paper did not reach 100%. This may indicate some limitations of the developed models that make it impossible to distinguish 'Debreceni botermo', 'Łutówka', 'Nefris', and 'Kelleris' sour cherry pits based on geometric features with 100% accuracy. It prompts us to carry out further research on sour cherry pits to build discriminative models combining selected geometric and other features. However, the contribution of this study to distinguishing sour cherry pit cultivars using machine learning is significant. The linear dimensions and shape factors with the highest discriminative power were indicated. The mean values of these selected parameters differed the most among the cultivars. The next stage of the research may involve combining these geometric features and selected textures in the model to increase the discrimination accuracy. The developed models based on geometric and textural features could be more successfully applied in practice to detect falsification of sour cherry pit cultivars.

## 4. Conclusions

The geometric parameters such as linear dimensions and shape factors proved to be useful for the discrimination of sour cherry pits belonging to different cultivars. Higher accuracies were observed when distinguishing pairs of pit cultivars than four cultivars. The discriminative models built based on sets of linear dimensions or shape factors and combined linear dimensions and shape factors provided very high results. However, the highest discriminative power for distinguishing the different cultivars of sour cherry pits was observed for combined linear dimensions and shape factors, whereas the linear dimensions were characterized by the lowest discriminative power. The present study was the first extensive approach to classify sour cherry pits belonging to different cultivars using innovative models built based on geometric features by machine learning algorithms. Such models developed using the sets of selected linear dimensions, shape factors and combined linear dimensions and shape factors for the discrimination of 'Debreceni botermo', 'Łutówka', 'Nefris', and 'Kelleris' sour cherry pits were not found in the available literature. The results of the discrimination based on geometric features were high, comparable to the results obtained for models built using texture parameters reported in previous studies. Demonstrating the usefulness of geometric features to distinguish sour cherry pit cultivars can have practical importance to authenticate pit samples and avoid mixing different cultivars with different chemical properties. However, the limitation of the proposed approach may be the accuracy of the discrimination, which was less than 100%. Therefore, future research may focus on developing the models combining the geometric and texture features to increase their discrimination accuracy.

**Author Contributions:** Conceptualization, E.R.; methodology, E.R.; software, E.R.; validation, E.R., K.S. and M.F.A.; formal analysis, E.R.; investigation, E.R.; resources, E.R.; data curation, E.R.; writing—original draft preparation, E.R., K.S. and M.F.A.; writing—review and editing, E.R., K.S., and M.F.A.; visualization, E.R.; supervision, E.R. All authors have read and agreed to the published version of the manuscript.

**Funding:** This research received no external funding.

**Institutional Review Board Statement:** Not applicable.

**Informed Consent Statement:** Not applicable.

**Data Availability Statement:** The data presented in this study are available on request from the corresponding author.

**Conflicts of Interest:** The authors declare no conflict of interest.

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
