# Peer review of "Discriminative Power of Geometric Parameters of Different Cultivars of Sour Cherry Pits Determined Using Machine Learning"

_agriculture, doi:10.3390/agriculture11121212_

Round 1

Reviewer 1 Report

For discrimination of sour cherry pits of different cultivars, k-fold cross validation should be considered.

Author Response

Reviewer 1

Comment: For discrimination of sour cherry pits of different cultivars, k-fold cross validation should be considered.

Answer: All test options available in WEKA software have been considered and tested. These test modes were:

  1. Use training set
  2. Supplied test set
  3. Cross-validation at folds: 10
  4. Percentage split

The appropriate test mode was selected on the basis of the obtained accuracies of discrimination and the experience gained during performing the previous studies.

The test option "1. Use training set" provided the highest results. However, it was not very useful and the most optimistic because it did not contain an independent test set that was not used for training. The test set and the training set were the same and all data were used to prepare the model and the same data were applied to evaluate the model. The results may be prone to overfitting.

The test option "2. Supplied test set" was very objective. The dataset was manually split without using WEKA software. Thus, the model was trained using training data (70%) and the dataset for testing (30%) was separate and included data that were not used for training.

As option "3. Cross-validation", the 10-fold cross-validation mode was selected. The dataset was randomly divided into 10 parts. Each part was held out in turn and regarded as the test set, and the remaining nine parts were treated as the training sets. The training procedure was executed a total of 10 times on different training sets. The average of 10 error estimates was determined as the overall error estimate. Ten folds are generally sufficient to obtain the best error estimate. However, this option required the creation of many more models and the cost including time of analysis may be higher.

The mode "4. Percentage split" included a random split of a dataset into a training and a testing set at a given percentage. It is similar to option "2. Supplied test set", fast and objective. However, the division of a dataset into training and testing sets was performed independently by the WEKA software without the possibility of controlling this stage.

The results were the highest for test option “1. Use training set". However, due to its least objectivity, it was not taken into account. For the other options, the results were quite similar. These observations confirmed the results of previous studies reported, e.g., by Ropelewska (2019) who found similar accuracies for 10-fold cross-validation and dataset split into training set: 70% and test set: 30%. The results for 10-fold cross-validation were slightly higher. However, the manual division of data was more objective and as much as 30% was used for testing.

Ropelewska E. 2019. Evaluation of wheat kernels infected by fungi of the genus Fusarium based on morphological features. J Food Saf. 39: e12623.

It has been more specified as follows:

“The criterion for evaluating the usefulness of datasets selected with the use of search methods was the highest correctness of discrimination. However, a great reduction in the number of parameters decreased the correctness of the discrimination and analyzes were performed with the exclusion of only a few attributes. The datasets were manually split into a training (70 %) and test set (30 %). The application of a separate test set that was not used for training ensured the objectivity of the results.”

Reviewer 2 Report

Dear editor

its sounds good, but references must be more around machine learning and to be compare by other researchers results especially about different plants, it must be improved.

it is interesting paper and new methods in statistics could be more comprehensive and the text is clear but need more references just about machine learning and its role in prediction also they do address the main question posed.

Regards

Author Response

Reviewer 2

Comment: Its sounds good, but references must be more around machine learning and to be compare by other researchers results especially about different plants, it must be improved.

It is interesting paper and new methods in statistics could be more comprehensive and the text is clear but need more references just about machine learning and its role in prediction also they do address the main question posed.

Answer: The authors are very grateful for this valuable comment. The additional references about machine learning have been cited. The Introduction section has been extended and supplemented with the following sentences:

“The application of machine learning may be useful for plant research. Machine learning as a sub-class of artificial intelligence is an important topic in the computer field. Currently, researchers strive to increase the precision of algorithms and the intelligence of machines. Learning became a significant part of machines. Due to computer vision, which is a domain of machine learning, machines can be trained for processing, analyzing, and recognizing visual data [14]. Machine learning is intended to enable machines to learn using the available data and make predictions. The learning of the computers automatically by themselves without human intervention may be important for precise prediction [15]. The prediction models developed using machine learning and artificial intelligence can provide promising and accurate results. The models based on artificial intelligence can learn from existing data and then predict even non-linear phenomena related to, e.g., prediction of food production, crop yield, identification of the number of immature fruits [16]. The application of machine learning in modern agriculture is important due to the increasing call for food, the necessity for increasing the effectiveness of agricultural practices and decreasing the environmental burden. Machine learning ensures an increase in computational power compared to conventional techniques of data processing, which can be incapable of extracting all necessary information from field data and thus meeting the growing demands of smart farming [17]. Machine learning focused on detection of disease, species and weeds in crops, prediction of crop yield and soil parameters, classification of crop images to evaluate the plant quality and yield can be one of the key components of the agricultural revolution [18].

In the case of the seed industry, machine learning may be important for the production, correct cultivar identification, identification of contaminations, quality control. The use of machine vision techniques can result in more accurate and faster classification results compared to the manual inspection performed by specialists based on the color and morphological features of seeds [19]. Machine learning caused significant advances in seed research by providing decision-making support and facilitating the development of robust approaches in the seed industry [20].”

  1. Sharma, N., Sharma, R., Jindal, N. Machine Learning and Deep Learning Applications-A Vision. Global Transitions Proceedings 2021, 2, 24-28.
  2. Asongo, A.I, Barma, M, Muazu, H.G. Machine Learning Techniques, methods and Algorithms: Conceptual and Practical Insights. International Journal of Engineering Research and Applications 2021, 11(8), 55-64.
  3. Nosratabadi, S., Ardabili, S., Lakner, Z., Mako, C., Mosavi, A. Prediction of Food Production Using Machine Learning Algo-rithms of Multilayer Perceptron and ANFIS. Agriculture 2021, 11(5), 408.
  4. Benos, L., Tagarakis, A.C., Dolias, G., Berruto, R., Kateris, D., Bochtis, D. Machine Learning in Agriculture: A Comprehensive Updated Review. Sensors 2021, 21, 3758.
  5. Sharma, A., Jain, A., Gupta, P., Chowdary, V. Machine learning applications for precision agriculture: A comprehensive review. IEEE Access 2020, 9, 4843-4873.
  6. Ajaz, R.H., Hussain, L. Seed Classification using Machine Learning Techniques. Journal of Multidisciplinary Engineering Science and Technology (JMEST) 2015, 2(5), 1098- 1102.
  7. de Medeiros, A.D., da Silva, L.J., Ribeiro, J.P.O., Ferreira, K.C., Rosas, J.T.F., Santos, A. A., da Silva, C. B. Machine learning for seed quality classification: An advanced approach using merger data from FT-NIR spectroscopy and X-ray imaging. Sensors 2020, 20(15), 4319.

Reviewer 3 Report

This article addresses a very interesting topic. The introduction is proper, and the statement of the problem is adequate. However, it is necessary the authors emphasize more on the contributions of this study both in the introduction section and in the discussion section. They should write one paragraph justifying why their approach is novel and why they think their solution for this research problem is different than others in the literature.

Meanwhile, important recent relevant literatures are missing, such as:

Nosratabadi, S., Ardabili, S., Lakner, Z., Mako, C., & Mosavi, A. (2021). Prediction of Food Production Using Machine Learning Algorithms of Multilayer Perceptron and ANFIS. Agriculture11(5), 408.

The conclusion section is very short, and it should be improved. The contributions should be discussed, and the findings should be compared with the others. Research limitations, recommendations for future studies, and the implementation of the results (i.e., who and how can benefit from the results) should be also mentioned in this section.  

Author Response

Reviewer 3

Comment 1: This article addresses a very interesting topic. The introduction is proper, and the statement of the problem is adequate. However, it is necessary the authors emphasize more on the contributions of this study both in the introduction section and in the discussion section. They should write one paragraph justifying why their approach is novel and why they think their solution for this research problem is different than others in the literature.

Answer: The authors are grateful for this comment. The introduction and the discussion have been supplemented. The paragraph providing the background related to seed classification and indicating the contribution and novelty of present research has been added to the Introduction as follows:

“In the case of the seed industry, machine learning may be important for the production, correct cultivar identification, identification of contaminations, quality control. The use of machine vision techniques can result in more accurate and faster classification results compared to the manual inspection performed by specialists based on the color and morphological features of seeds [19]. Machine learning caused significant advances in seed research by providing decision-making support and facilitating the development of robust approaches in the seed industry [20]. The usefulness of the application of machine learning for seed classification was reported in the available literature. The machine learning models were built based on various image features. In the case of cultivar discrimination of fruit seeds or pits and stones, high efficiency of models based on texture parameters was reported for pepper seeds [21], apple seeds [22], peach seeds and stones [23], sour cherry pits [24] and sweet cherry pits [25]. Furthermore, the geometric features proved to be useful for the pit or stone discrimination for different cultivars of apricot [26], plum [27,28,29], olive [30], jujube [31], sweet cherry [25]. However, in the present study, such extensive research using dozens of geometric parameters including linear dimensions and shape factors was performed for the first time to discriminate sour cherry pits ‘Debreceni botermo’, ‘Łutówka’, ‘Nefris’, ‘Kelleris’ using different classifiers (machine learning algorithms). The innovative models based on the sets of selected linear dimensions, shape factors and combined linear dimensions and shape factors were developed. This approach to distinguishing cultivars of sour cherry pits was original.”

  1. Ajaz, R.H., Hussain, L. Seed Classification using Machine Learning Techniques. Journal of Multidisciplinary Engineering Science and Technology (JMEST) 2015, 2(5), 1098- 1102.
  2. de Medeiros, A.D., da Silva, L.J., Ribeiro, J.P.O., Ferreira, K.C., Rosas, J.T.F., Santos, A. A., da Silva, C. B. Machine learning for seed quality classification: An advanced approach using merger data from FT-NIR spectroscopy and X-ray imaging. Sensors 2020, 20(15), 4319.
  3. Ropelewska, E., Szwejda-Grzybowska, J. A comparative analysis of the discrimination of pepper (Capsicum annuum L.) based on the cross‐section and seed textures determined using image processing. Journal of Food Process Engineering 2021, 44, 13694.
  4. Ropelewska, E. The use of seed texture features for discriminating different cultivars of stored apples. Journal of Stored Products Research 2020, 88, 101668.
  5. Ropelewska, E., Rutkowski, K.P. Differentiation of peach cultivars by image analysis based on the skin, flesh, stone and seed textures. European Food Research and Technology 2021, 247, 2371-2377.
  6. Ropelewska, E. Classification of the pits of different sour cherry cultivars based on the surface textural features. Journal of the Saudi Society of Agricultural Sciences 2021, 20, 52-57.
  7. Ropelewska, E. The Application of Machine Learning for Cultivar Discrimination of Sweet Cherry Endocarp. Agriculture 2021, 11, 6.
  8. Milatović, D., Ðurović, D., Milivojević, J. Stone and kernel characteristics as elements in identification of apricot cultivars. Voćarstvo 2006, 40, 311-319.
  9. Depypere, L., Chaerle, P., Mijnsbrugge, K.V., Goetghebeur, P. Stony endocarp dimension and shape variation in Prunus section Prunus. Ann. Bot. 2007, 100, 1585-1597.
  10. Sarigu, M., Grillo, O., Lo Bianco, M., Ucchesu, M., d’Hallewin, G., Loi, M.C., Venora, G., Bacchetta, G. Phenotypic identification of plum varieties (Prunus domestica L.) by endocarps morpho-colorimetric and textural descriptors. Comput. Electron. Agric. 2017, 136, 25-30.
  11. Frigau, L., Antoch, J., Bacchetta, G., Sarigu, M., Ucchesu, M., Alves, Ch.Z., Mola, F.A. Statistical Approach to the Morpho-logical Classification of Prunus sp. Seeds. Plant Biosyst. 2020, 154, 1-16.
  12. Beyaz, A., Öztürk, R. Identification of olive cultivars using image processing techniques. Turk. J. Agric. For. 2016, 40, 671-683.
  13. Kim, S.H., Nam, J.I., Kim, C.W. Analysis of Qualitative and Quantitative Traits to Identify Different Chinese Jujube Cultivars. Plant Breed. Biotechnol. 2019, 7, 175-185.

The Discussion section has been supplemented with the following sentences:

“The results of cultivar discrimination of sour cherry pits based on geometric parameters presented in this paper did not reach 100 %. This may indicate some limitations of developed models that make it impossible to distinguish sour cherry pits ‘Debreceni botermo’, ‘Łutówka’, ‘Nefris’, ‘Kelleris’ based on geometric features with 100 % probability. It prompts to carry out further research for sour cherry pits to build discriminative models combining selected geometric and other features. However, the contribution of this study to distinguishing sour cherry pit cultivars using machine learning is significant. The linear dimensions and shape factors with the highest discriminative power were indicated. The mean values of these selected parameters differed the most among the cultivars. The next stage of the research may involve combining these geometric features and selected textures in the model to increase the discrimination accuracy. The developed models based on geometric and textural features could be more successfully applied in practice to detect falsification of sour cherry pit cultivars.”

Comment 2: Meanwhile, important recent relevant literatures are missing, such as:

Nosratabadi, S., Ardabili, S., Lakner, Z., Mako, C., & Mosavi, A. (2021). Prediction of Food Production Using Machine Learning Algorithms of Multilayer Perceptron and ANFIS. Agriculture11(5), 408.

Answer: The authors are very grateful for this valuable comment. The suggested literature is very valuable. Adding this and other references significantly improved the manuscript. The following sentences and references have been added:

“The application of machine learning may be useful for plant research. Machine learning as a sub-class of artificial intelligence is an important topic in the computer field. Currently, researchers strive to increase the precision of algorithms and the intelligence of machines. Learning became a significant part of machines. Due to computer vision, which is a domain of machine learning, machines can be trained for processing, analyzing, and recognizing visual data [14]. Machine learning is intended to enable machines to learn using the available data and make predictions. The learning of the computers automatically by themselves without human intervention may be important for precise prediction [15]. The prediction models developed using machine learning and artificial intelligence can provide promising and accurate results. The models based on artificial intelligence can learn from existing data and then predict even non-linear phenomena related to, e.g., prediction of food production, crop yield, identification of the number of immature fruits [16]. The application of machine learning in modern agriculture is important due to the increasing call for food, the necessity for increasing the effectiveness of agricultural practices and decreasing the environmental burden. Machine learning ensures an increase in computational power compared to conventional techniques of data processing, which can be incapable of extracting all necessary information from field data and thus meeting the growing demands of smart farming [17]. Machine learning focused on detection of disease, species and weeds in crops, prediction of crop yield and soil parameters, classification of crop images to evaluate the plant quality and yield can be one of the key components of the agricultural revolution [18].

  1. Sharma, N., Sharma, R., Jindal, N. Machine Learning and Deep Learning Applications-A Vision. Global Transitions Proceedings 2021, 2, 24-28.
  2. Asongo, A.I, Barma, M, Muazu, H.G. Machine Learning Techniques, methods and Algorithms: Conceptual and Practical Insights. International Journal of Engineering Research and Applications 2021, 11(8), 55-64.
  3. Nosratabadi, S., Ardabili, S., Lakner, Z., Mako, C., Mosavi, A. Prediction of Food Production Using Machine Learning Algorithms of Multilayer Perceptron and ANFIS. Agriculture 2021, 11(5), 408.
  4. Benos, L., Tagarakis, A.C., Dolias, G., Berruto, R., Kateris, D., Bochtis, D. Machine Learning in Agriculture: A Comprehensive Updated Review. Sensors 2021, 21, 3758.
  5. Sharma, A., Jain, A., Gupta, P., Chowdary, V. Machine learning applications for precision agriculture: A comprehensive review. IEEE Access 2020, 9, 4843-4873.

Comment 3: The conclusion section is very short, and it should be improved. The contributions should be discussed, and the findings should be compared with the others. Research limitations, recommendations for future studies, and the implementation of the results (i.e., who and how can benefit from the results) should be also mentioned in this section.  

Answer: The authors fully agree with this suggestion. The conclusion section has been extended and supplemented with information mentioned by the Reviewer as follows:

“The present study was the first extensive approach to classify sour cherry pits belonging to different cultivars using innovative models built based on geometric features by machine learning algorithms. Such models developed using the sets of selected linear dimensions, shape factors and combined linear dimensions and shape factors for the discrimination sour cherry pits ‘Debreceni botermo’, ‘Łutówka’, ‘Nefris’, ‘Kelleris’ were not found in the available literature. The results of discrimination based on geometric features were high, comparable to the results obtained for models built using texture parameters reported in the previous studies. Demonstrating the usefulness of geometric features to distinguish sour cherry pit cultivars can have practical importance to authenticate the pit samples and avoid mixing different cultivars with different chemical properties. However, the limitation of the proposed approach may be the accuracy of the discrimination, which was less than 100 %. Therefore, future research may focus on developing the models combining the geometric and texture features to increase the discrimination accuracy.”

Reviewer 4 Report

The paper presents problem related to using machine learning for determining discriminative power of geometric parameters of different cultivars of sour cherry pits. This is important problem from a food production management point of view. This is originality and scientific contributions on considered research area. The paper is relevant to Agriculture topics. The language and structure for a scientific paper is proper.

Recommendations to the authors for improving their paper:

The contribution of the research should be clearly included in the “Introduction” section.

The sentence: “Therefore, correct recognition of the cultivar may be important for food processing, and it may have a significant impact on the properties of the final product” is very general. The “food processing” and “impact on the  properties of the final product” should be explained more detailed.

Conclusions:

what are the disadvantages/limitations of proposed approach? what are the implications of proposed approach for science and practice? Authors should indicate the future research works.

Author Response

Reviewer 4

The paper presents problem related to using machine learning for determining discriminative power of geometric parameters of different cultivars of sour cherry pits. This is important problem from a food production management point of view. This is originality and scientific contributions on considered research area. The paper is relevant to Agriculture topics. The language and structure for a scientific paper is proper.

Answer: The authors are grateful for this comment.

Recommendations to the authors for improving their paper:

Comment 1: The contribution of the research should be clearly included in the “Introduction” section.

Answer: The authors are grateful for this suggestion. The Introduction has been supplemented with the following sentences which provide the background related to seed classification and indicated the contribution of present research:

“In the case of the seed industry, machine learning may be important for the production, correct cultivar identification, identification of contaminations, quality control. The use of machine vision techniques can result in more accurate and faster classification results compared to the manual inspection performed by specialists based on the color and morphological features of seeds [19]. Machine learning caused significant advances in seed research by providing decision-making support and facilitating the development of robust approaches in the seed industry [20]. The usefulness of the application of machine learning for seed classification was reported in the available literature. The machine learning models were built based on various image features. In the case of cultivar discrimination of fruit seeds or pits and stones, high efficiency of models based on texture parameters was reported for pepper seeds [21], apple seeds [22], peach seeds and stones [23], sour cherry pits [24] and sweet cherry pits [25]. Furthermore, the geometric features proved to be useful for the pit or stone discrimination for different cultivars of apricot [26], plum [27,28,29], olive [30], jujube [31], sweet cherry [25]. However, in the present study, such extensive research using dozens of geometric parameters including linear dimensions and shape factors was performed for the first time to discriminate sour cherry pits ‘Debreceni botermo’, ‘Łutówka’, ‘Nefris’, ‘Kelleris’ using different classifiers (machine learning algorithms). The innovative models based on the sets of selected linear dimensions, shape factors and combined linear dimensions and shape factors were developed. This approach to distinguishing cultivars of sour cherry pits was original.”

  1. Ajaz, R.H., Hussain, L. Seed Classification using Machine Learning Techniques. Journal of Multidisciplinary Engineering Science and Technology (JMEST) 2015, 2(5), 1098- 1102.
  2. de Medeiros, A.D., da Silva, L.J., Ribeiro, J.P.O., Ferreira, K.C., Rosas, J.T.F., Santos, A. A., da Silva, C. B. Machine learning for seed quality classification: An advanced approach using merger data from FT-NIR spectroscopy and X-ray imaging. Sensors 2020, 20(15), 4319.
  3. Ropelewska, E., Szwejda-Grzybowska, J. A comparative analysis of the discrimination of pepper (Capsicum annuum L.) based on the cross‐section and seed textures determined using image processing. Journal of Food Process Engineering 2021, 44, 13694.
  4. Ropelewska, E. The use of seed texture features for discriminating different cultivars of stored apples. Journal of Stored Products Research 2020, 88, 101668.
  5. Ropelewska, E., Rutkowski, K.P. Differentiation of peach cultivars by image analysis based on the skin, flesh, stone and seed textures. European Food Research and Technology 2021, 247, 2371-2377.
  6. Ropelewska, E. Classification of the pits of different sour cherry cultivars based on the surface textural features. Journal of the Saudi Society of Agricultural Sciences 2021, 20, 52-57.
  7. Ropelewska, E. The Application of Machine Learning for Cultivar Discrimination of Sweet Cherry Endocarp. Agriculture 2021, 11, 6.
  8. Milatović, D., Ðurović, D., Milivojević, J. Stone and kernel characteristics as elements in identification of apricot cultivars. Voćarstvo 2006, 40, 311-319.
  9. Depypere, L., Chaerle, P., Mijnsbrugge, K.V., Goetghebeur, P. Stony endocarp dimension and shape variation in Prunus section Prunus. Ann. Bot. 2007, 100, 1585-1597.
  10. Sarigu, M., Grillo, O., Lo Bianco, M., Ucchesu, M., d’Hallewin, G., Loi, M.C., Venora, G., Bacchetta, G. Phenotypic identification of plum varieties (Prunus domestica L.) by endocarps morpho-colorimetric and textural descriptors. Comput. Electron. Agric. 2017, 136, 25-30.
  11. Frigau, L., Antoch, J., Bacchetta, G., Sarigu, M., Ucchesu, M., Alves, Ch.Z., Mola, F.A. Statistical Approach to the Morpho-logical Classification of Prunus sp. Seeds. Plant Biosyst. 2020, 154, 1-16.
  12. Beyaz, A., Öztürk, R. Identification of olive cultivars using image processing techniques. Turk. J. Agric. For. 2016, 40, 671-683.
  13. Kim, S.H., Nam, J.I., Kim, C.W. Analysis of Qualitative and Quantitative Traits to Identify Different Chinese Jujube Cultivars. Plant Breed. Biotechnol. 2019, 7, 175-185.

Comment 2: The sentence: “Therefore, correct recognition of the cultivar may be important for food processing, and it may have a significant impact on the properties of the final product” is very general. The “food processing” and “impact on the properties of the final product” should be explained more detailed.

Answer: The sentence has been rewritten as follows:

“Due to the dependence of the chemical properties of sour cherry kernels on the cultivar, correct cultivar recognition may be important in practice. The processing of cherry kernels may require a uniform sample of kernels with the same characteristics. Some cultivars with certain chemical properties may be more desirable for processing than others. Therefore, there may be a need for authentication to avoid adulteration and mixing different cultivars.”

Comment 3: Conclusions:

what are the disadvantages/limitations of proposed approach? what are the implications of proposed approach for science and practice? Authors should indicate the future research works.

Answer: The authors are very grateful for this comment. The Conclusions section has been extended and supplemented as follows:

“The present study was the first extensive approach to classify sour cherry pits belonging to different cultivars using innovative models built based on geometric features by machine learning algorithms. Such models developed using the sets of selected linear dimensions, shape factors and combined linear dimensions and shape factors for the discrimination sour cherry pits ‘Debreceni botermo’, ‘Łutówka’, ‘Nefris’, ‘Kelleris’ were not found in the available literature. The results of discrimination based on geometric features were high, comparable to the results obtained for models built using texture parameters reported in the previous studies. Demonstrating the usefulness of geometric features to distinguish sour cherry pit cultivars can have practical importance to authenticate the pit samples and avoid mixing different cultivars with different chemical properties. However, the limitation of the proposed approach may be the accuracy of the discrimination, which was less than 100 %. Therefore, future research may focus on developing the models combining the geometric and texture features to increase the discrimination accuracy.”
